# An Examination of an Iconic Trap-Neuter-Return Program: The Newburyport, Massachusetts Case Study

**DOI:** 10.3390/ani7110081

**Published:** 2017-10-31

**Authors:** Daniel D. Spehar, Peter J. Wolf

**Affiliations:** 1Independent Researcher, 4758 Ridge Road, #409, Cleveland, OH 44144, USA; danspehar9@gmail.com; 2Best Friends Animal Society, 5001 Angel Canyon Road, Kanab, UT 84741, USA

**Keywords:** free-roaming cats, feral cats, stray cats, trap-neuter-return, TNR, sterilization

## Abstract

**Simple Summary:**

Local communities in the United States are commonly responsible for selecting the most appropriate method of managing free-roaming cats. Lethal management has been widely utilized for generations, but the use of trap-neuter-return (TNR) has grown in recent decades. Despite expanded use of TNR, a relative scarcity of data associated with such programs exists. This paper retrospectively examines an iconic TNR program—began in 1992—that resulted in the elimination of hundreds of cats from the Newburyport, Massachusetts, waterfront. A careful review of contemporaneous reports, extant program documents, and stakeholder testimony indicates that an estimated 300 cats resided in the area at the commencement of the TNR program; none remained 17 years later. Up to one-third of the cats trapped were sociable and adopted into homes; the remainder were sterilized and vaccinated before being returned to the waterfront, where they declined in number over time due to attrition. A compelling narrative emerged from the available evidence concerning the effectiveness of TNR as a management practice, although a lack of feline population data associated with the Newburyport TNR program underscores the need for establishment of standardized data collection and assessment practices.

**Abstract:**

The use of trap-neuter-return (TNR) as a humane alternative to the lethal management of free-roaming cats has been on the rise for several decades in the United States; however a relative paucity of data from TNR programs exists. An iconic community-wide TNR effort; initiated in 1992 and renowned for having eliminated hundreds of free-roaming cats from the Newburyport; Massachusetts waterfront; is cited repeatedly; yet few details appear in the literature. Although the presence of feline population data was quite limited; a detailed narrative emerged from an examination of contemporaneous reports; extant TNR program documents; and stakeholder testimony. Available evidence indicates that an estimated 300 free-roaming cats were essentially unmanaged prior to the commencement of the TNR program; a quick reduction of up to one-third of the cats on the waterfront was attributed to the adoption of sociable cats and kittens; the elimination of the remaining population; over a 17-year period; was ascribed to attrition. These findings illuminate the potential effectiveness of TNR as a management practice; as well as call attention to the need for broad adoption of systematic data collection and assessment protocols.

## 1. Introduction

An air of ambiguity permeates most candid discussions of free-roaming cats and their management. Facts as fundamental as the number of unowned, free-roaming cats in the U.S. are elusive; estimates vary greatly, from 32 million [1] to a sum approaching the nation’s owned cat population [2,3,4], which is likely between 74 million [5] and 94.2 million [6]. No matter the actual population of free-roaming cats nationwide, local communities are most often faced with the challenge of determining how best to control their numbers. Ultimately, such management is reduced to a choice between lethal and non-lethal methods. Historically, lethal approaches have been utilized most often [7,8,9,10]. Today, this generally takes the form of complaint-based shelter impoundment followed by lethal injection. Although this approach has been used for generations in the U.S., there is no research to suggest that it is effective for the reduction of free-roaming cats in a community. However, research studies have demonstrated the ability of intensive, long-term eradication campaigns to eliminate free-roaming cat populations on a number of oceanic islands [11,12].

Increased advocacy on behalf of free-roaming cats over the past quarter-century has resulted in an increase in the practice of trap-neuter-return (TNR) as a humane alternative to lethal management [7,13,14,15,16,17]. However, a relative lack of systematically collected data has led some conservationists to question the effectiveness of these programs [7,18,19,20]. Multiple research studies have documented site-specific declines in free-roaming cat populations as a result of TNR [4,21,22,23], including the elimination of individual colonies [24] and reduction [25] or elimination [26] of kitten births. Moreover, significant declines in feline shelter intake and euthanasia have been observed in locations where high-impact targeted TNR [27] and community-wide return to field [28] programs have been implemented. Nevertheless, it has been suggested that an “information vacuum” exists relative to the innumerable TNR programs carried out across the U.S. over the past 25 years [7] (p. 1). Because robust data from these programs have been scarce, determinations about program impacts have typically been based on anecdotal evidence [7,18,24].

Recently, some have begun calling for more standardized data collection and assessment practices (drawn largely from the fields of biology and wildlife management) to better understand the effectiveness of TNR programs [7,10,18]. The objective of this initiative is to establish greater uniformity in TNR processes and practices so that systematic tracking and assessment of free-roaming cat populations can be performed consistently across varying contexts [7,18]. It has been suggested that widespread adherence to such a protocol would lead to more efficient and effective non-lethal management practices and “provide a basis for constructive engagement [among stakeholder groups] about cat management issues” [7] (p. 1).

To reach an informed consensus among free-roaming cat management stakeholders, it is important to compile quantitative assessments derived from TNR program data that have been recorded consistently over time. Nevertheless, until such time as TNR program data collection and assessment practices have been standardized, it is useful to ascertain as much information as possible from existing examples of community-based TNR programs. Despite the potential deficiencies of such data, insights acquired from decades of extensive TNR field experience are likely to prove valuable to those interested in better understanding this topic. 

It is in this vein that the present historical case study, which examines one of the most well-known and longest-running TNR campaigns in the U.S., was undertaken. The TNR program was begun in 1992 by the Merrimack River Feline Rescue Society (MRFRS) on the central waterfront of Newburyport, Massachusetts and has since been widely cited as an example of TNR success on a community level [29,30,31]; however, to date, only superficial reports about what took place have been published. The present study is a qualitative, descriptive analysis of the available evidence documenting the actions and conditions that led to the reported elimination of an estimated 300 free-roaming cats from the area adjacent to the Merrimack River in Newburyport over a 17-year period.

## 2. Materials and Methods 

### 2.1. Interviews 

As this was a retrospective study of an impromptu TNR program initiated a quarter-century ago by citizen volunteers for the sole purpose of reducing a large local free-roaming cat population, it was not surprising to discover early on in the investigation that the existence of statistical data was very limited. Notwithstanding the scarcity of systematically collected numeric data, much other evidence relating to Newburyport’s waterfront cat management efforts was collected and reviewed. Correspondence via phone interviews and email with former and current MRFRS volunteers and local government office-holders, newspaper accounts, and surviving documents (e.g., meeting minutes), were examined in an attempt to uncover and substantiate the sequence of events.

A semi-structured interview format featuring open-ended questions was utilized for individual interviews. Suggestive questioning [32] was avoided in order to allow interviewed stakeholders to elaborate on situational details completely from their own recollections. Group email threads were employed for clarification of specific points and in order to fill information gaps. Triangulation of sourcing was employed whenever possible; in addition, corroboration from multiple sources was obtained for key information uncovered exclusively via the interview process, in an attempt to compensate for variations in perception and potential deficiencies in individual memory. 

Limited storage space prevented MRFRS from archiving many older records (e.g., medical/sterilization records and feeding logs); therefore, almost none of this type of evidence was available for review [33]. Similarly, vaccination records for cats who passed through the program were not obtainable. Veterinarians in Massachusetts are required to keep copies of such records for four years [34], and municipalities for two years [35]; hence, vaccination records could not be produced by the veterinarians who performed TNR services [36,37] or by the clerk’s office for the city of Newburyport [38]. Together, this lack of records meant that a quantitative statistical analysis was not possible; nevertheless, a substantive narrative emerged from a careful review of the available evidence.

### 2.2. Site Description and Program Origins

Newburyport, Massachusetts is a coastal town located at the mouth of the Merrimack River, near the Atlantic Ocean; approximately 56 km northeast of Boston. Newburyport’s human population in 1990 was 16,324 [39]. The site of the TNR program (Figure 1) documented here was approximately 1.9 km in length and a 0.4 km wide moving inland from the river [40,41]. Then, as now, the central waterfront was bisected by a highway/bridge (Rt. 1) that crossed the river northward into Salisbury, MA. The area east of Rt. 1 consisted of a light commercial district made up of shops, restaurants, museums, and theaters—with some residential property above—as well as single-family residences, boatyards, a vacant tannery building, and a U.S. Coast Guard station [33,42,43]; the area west of Rt. 1 was predominantly made up of residences, open parkland, boat docks, and a shuttered silversmithing plant [42,44]. By the early 1990s, many of Newburyport’s once-busy textile mills and other manufacturing sites along the river had been replaced by tourist-friendly businesses and attractions as part of community revitalization efforts that began decades earlier [42,43,45]. It was estimated that three quarters of the cats living on the waterfront prior to inception of the TNR program inhabited the area east of the highway, closer to restaurants and other easily-accessible sources of food [46,47,48].

Ultimately, it was concern about hungry cats boldly scavenging for food in full view of downtown restaurant patrons that became the impetus for starting the TNR program [49,50,51]. Two concerned Newburyport residents, Dorothy Fairweather and Jan DeWitt, independently approached the president of the local chamber of commerce, Shirley Magnanti, about enlisting support from local businesses in implementing a humane solution to address the free-roaming cat problem. Fairweather and DeWitt were advised to collaborate and form a citizen committee, while Magnanti went about convincing the town’s business owners to back their plan to trap, sterilize and return cats to the waterfront [50,52,53]. Aside from a few initial skeptics, support from the business community came quickly, recalled Magnanti [53]. The cat problem was decades old, so awareness existed. The challenge, according to Magnanti, was persuading local restaurateurs and shopkeepers that TNR was the most appropriate way to address the overpopulation of free-roaming cats. “I simply made an appeal to local business owners that it would be bad for business if Newburyport was known as a place that killed cats” [53].

Within weeks, in June of 1992, the Newburyport TNR initiative was formally established by a committee of 11 citizens as the “Newburyport Neuter and Release Program” [54] (soon thereafter, the group’s name was changed to the Merrimack River Feline Rescue Society [55]). Fairweather was named the group’s first president, a title she would hold for approximately five years [52]. The group’s mission was to solve the increasingly noticeable, yet decades-old, “serious civic problem” caused by the large numbers of stray and feral cats living in close proximity to downtown restaurants and other businesses [51] (p. 5). Contemporaneous reports confirmed that large numbers of free-roaming cats in the area fed on scraps left by restaurant-goers and staff, as well as whatever else they could scavenge; nevertheless, some appeared to be malnourished [56,57,58]. Moreover, it was common in early spring for boatyard workers to find dead cats who had sought winter cover in boats moored at the docks [50,51]. Many local business owners had expressed concern that the presence of cats begging for food and congregating around garbage dumpsters was having a negative impact on their businesses, and some were desperate for a solution to the problem [50,51]. “Newburyport has so many restaurants. Every place you go there are restaurants, there are dumpsters, and there are wild cats”, explained Carol LaRocque, municipal animal control officer during that period [56]. It was rumored that in the months just prior to implementation of the TNR program, attempts were made by a small number of local business owners to poison the cats [50,59,60,61]. The accusations of poisoning were never confirmed, according to Steve Fram, Newburyport’s director of public health at the time [62]. Yet even as some merchants and restaurant owners simply “wanted the cats gone” [61], others valued them as effective rodent deterrents and continued to feed them even while acknowledging that their numbers had grown too large [50,62]. It has been widely reported [29,30,31,56,63] and long-standing “tribal knowledge” among Newburyport’s residents, according to Stacy LeBaron, who succeeded Fairweather as MRFRS president, that hundreds of free-roaming cats lived on the town’s central waterfront when the TNR program began in mid-1992 [64].

“I do think that between 300 and 400 was the initial count of ferals, strays, and community cats in the entire waterfront area”, explained Patte Grimes, early MRFRS board member and cat trapper. “That includes downtown areas and the full stretch of the waterfront. The cats hung out where there were dumpsters at restaurants, and also at boatyards where fishermen would throw them pieces of fish they cleaned. That’s why [the area] could sustain that number, even considering there was a very high mortality rate of kittens and sicker adult cats before the early MRFRS founders started this process”, she said [65]. Attempts were made to more-precisely quantify the number of cats. Although conducted in an informal manner, founding MRFRS members, Nancy and Bob MacNeil, took a census of all the cats living on the waterfront. The MacNeils were described by Fairweather as “champion trappers, [who] knew every cat on the waterfront” [66]. The couple spent many hours near the banks of the river capturing as well as caring for the cats; they also fostered in their home many of the cats and kittens held for adoption. “I counted the cats with my husband as best we could”, Nancy MacNeil explained. “I think that [300] is the number” [67]. Another early MRFRS volunteer, Jerry Mullins, recalled a separate effort to inventory the cats in each of the colonies (which peaked in number at 14), which he estimated “averaged around 20” apiece [68]. Regrettably, no written records of these attempts to count the number of cats on the waterfront in the early 1990s are extant.

## 3. Results

### 3.1. Implementation of Newburyport’s Waterfront TNR Program

Within days of the group’s forming, volunteers began trapping free-roaming cats on the waterfront for the purpose of having them spayed or neutered, vaccinated for rabies and panleukopenia, tested for feline immunodeficiency virus (FIV) and feline leukemia (FeLV), tattooed on the right ear (as a means of identification prior to the advent of microchipping) and notched on the left ear (in order to signify that a cat had been sterilized—this practice was eventually replaced by ear-tipping), prior to cats being returned to locations of capture [50,58]. Cats were typically taken to one of two local veterinary clinics for sterilization. The veterinarians who operated each of these clinics, Regina Downey and John Grillo, also served on the group’s board of directors and offered reduced pricing [50,52,69]. In the early years of the program, cats who tested positive for FIV or FeLV were humanely euthanized. This practice, along with routine FIV/FeLV testing itself, was discontinued in 1998 [36,50,52,70].

Consistent with the findings of Tan et al. [23], multiple volunteer trappers claimed that it was common for at least half of the cats at new trapping sites to be caught within two trapping nights, although it was acknowledged that the rate of success often declined after the initial round of trapping [67,71]. A similar increase in required trapping effort as the proportion of non-sterilized cats in a colony declines was observed by Nutter [24] and has been demonstrated in cat population management modelling [10]. Grimes explained the process: “The more cats you’re going after, the higher your success rate. Where it gets tricky is when you’ve trapped 23 out of 25 of the cats in the colony, and there are always 1 or 2 that are smart and savvy, and won’t go in. That’s when persistence and patience pays off. We didn’t have drop traps back then like we do now to get the hard ones. If it gets to the point where no one else is going in the traps, you pull everything and wait a week or so, so they relax and things get back to normal, then, try again” [71].

MRFRS volunteers reported experiencing high levels of trapping success, eventually resulting in 100% of waterfront cats being sterilized. Similar levels of trapping success have been observed at sites elsewhere [22,24] and have been attributed to the use of regular feeding times and locations [23]. The withholding of food from targeted cats for at least 24 hours before trapping was also of vital importance, according to Grimes [71].

Kittens born on the waterfront (before all adult cats had been sterilized) were trapped for adoption as soon as possible after their discovery [71,72]. A similar protocol was observed by Natoli et al. [21] in the management of Rome’s feral cat colonies. In the early years of the Newburyport program, pregnant females who were trapped were allowed to deliver their kittens in foster care before being sterilized and returned or adopted; later, a decision was made to terminate the pregnancies of expectant mother cats. Due to these practices, it was believed that, as in Rome [21], kitten births were at most an incidental contributor to the population of free-roaming cats after initiation of the TNR program [72,73].

Volunteers fed the hundreds of cats on the waterfront twice per day [50,58,61,74]. They maintained feeding logs to track the amount of food consumed, individual cats observed, and any new or unusual circumstances, such as new arrivals or suspected health problems [50,68]. Feeding logs were kept on site (on clipboards stored inside the feeding stations) and updated by colony caretakers at each visit. Newly arriving cats, whether due to abandonment or migration, were subjected to the same TNR process described above [52,75]. In addition, regular “feral feeder” meetings were held to facilitate communication among caretakers and to promptly address issues that arose at colony sites [50,75,76]. At the height of the program, a total of 14 feeding stations were installed at discrete locations across the waterfront [50,77]. It is estimated that most stations were spaced between 140 and 200 meters apart, although interviewees suggested that several were likely situated more closely for topographical reasons (Figure 1). The number of active feeding stations varied as the composition of the waterfront’s free-roaming cat population changed over time [78,79].

Many cats were regularly observed at specific feeding sites, although some cats fed at multiple locations [68,75]. Nutter [24] noted that regular observation of colony members by caretakers allows for easier identification of irregularities, such as health problems and new arrivals, and results in greater accessibility for management activities.

Almost immediately upon program inception, it became apparent to the group that not all the cats on the waterfront were “feral”, as had been originally assumed. “We had planned on [doing only] TNR and then we ran into “nice” cats. We were initially stumped as to what to do”, explained MRFRS co-founder and colony caregiver, Sheila Mullins [68]. A decision was made that group members would foster sociable cats until permanent homes could be found. Many of the waterfront cats were, in fact, sociable, so the number of cats being fostered grew quickly [50,61,74]. By the end of 1993, as MRFRS expanded its mission to include rescuing cats from nearby communities, the number of adoptable cats and kittens requiring housing became too great for the foster network to handle alone; thus, a permanent shelter space was opened above the clinic operated by Dr. Downey in the neighboring community of Salisbury [50,77,80]. “The original shelter was opened out of necessity”, explained Fairweather. “We had lots of cats and kittens who were not feral (i.e., strays and drop-offs) and we were keeping them in our homes until we (the volunteers) could not take in any more. I, as president, could no longer monitor the conditions or the health of the cats in the various homes. We needed a shelter/adoption center where the public was welcome and where we had control of the cleanliness and appearance of the space” [48].

Although operating a limited-admission shelter was not part of the group’s original mission, multiple MRFRS volunteers believed that opening the facility played an important role in mitigating what had been a significant source of cats on the waterfront by providing the town’s residents with an alternative to abandonment [47,48,53,81]. Within a year of the shelter’s opening, the waterfront had become only an incidental source of admissions to the facility, due apparently to the effects of the ongoing TNR campaign [47,81].

In addition to the unexpected need to house significant numbers of adoptable cats, MRFRS was forced to overcome several other challenges early on. The first of these was that many local residents were unfamiliar with the concept of returning recently-captured and sterilized cats to the location from which they were trapped. “Some people thought we were crazy putting the cats back after just trapping them”, explained Grimes [82]. Efforts to inform the public began almost immediately and persisted for the duration of the waterfront program. Educational tactics included regular tabling at community events, soliciting media coverage, and imaginative community outreach [48,81]. Fairweather explained: “The first weekend after our very first meeting as a committee, we set up a table at the festival that was going on in Newburyport. Since we started trapping immediately after we formed our organization, the local press soon started to publicize our activities. After we moved into the first shelter/adoption center, we brought in groups of young people (e.g., Brownie and Girl Scout troops) for tours of the shelter and educated them on our mission to reduce the numbers of kittens and homeless cats. We also brought some of our special cats to nursing homes to engage residents. Some patients who didn’t respond to any humans would smile as they were holding the cats” [48].

Outreach efforts were not limited to Newburyport; MRFRS assisted neighboring towns in establishing their own TNR programs. “From day one, we were willing to help others the best way we could at starting TNR programs. So, we started mentoring really even before we were successful on the waterfront, but it was a testament to the fact that we believed in our solution”, explained LeBaron [81].

Another, perhaps more formidable early challenge was noted by Fairweather: “The MRFRS founders started from scratch” [74]. In the early 1990s, very few TNR programs or protocols were available for the group to emulate. Additionally, in this pre-internet era, group leaders had very little access to information about other TNR programs being formed elsewhere [74]. Fairweather recalled that her introduction to TNR came by reading a blurb in an animal welfare magazine. Shortly thereafter, she came across and purchased a video (no longer in her possession) about a TNR program in the United Kingdom, which eventually became her inspiration for helping initiate the Newburyport waterfront program [52].

### 3.2. Timeline of Population Reduction

The following is an approximate timeline, assembled from meeting minutes, newspaper accounts, and stakeholder input, describing the decline in free-roaming cat population on the Newburyport waterfront:The first cats were trapped in June of 1992 on the property of the Captain’s Quarters restaurant (Figure 1, site #6) [48,54].“In the [first] two weeks, five cats were brought to Coastal Animal Clinic. Four were neutered and released, one had to be euthanized due to [FIV]” [59].A month after inception, 18 cats had been trapped and sterilized under the nascent program. Three adult males were euthanized due to viruses. Kittens and sociable adults were removed and put up for adoption [83].By the end of 1992, it was estimated that all but 100 cats originally living on the waterfront had been trapped, sterilized, and returned or made available for adoption [67,84]. “The numbers of cats started to decline very quickly”, MacNeil recalled [67]. According to Fairweather, fewer cats could be observed “almost immediately” because of the removal for adoption of sociable cats and socializable kittens [52].December 12, 1993: the MRFRS cats-only limited admission shelter, located above the Coastal Animal Clinic in Salisbury, officially opened [50,80,85].December 1995: it was reported in the *Boston Globe*’s North Weekly that “about 200” cats continued to reside on Newburyport’s waterfront [49] (p. 6).In the late 1990s two colonies, consisting of more than 20 cats in total, were removed due to building construction [44] and colony caretaker issues. Some of the cats were placed in indoor homes, while 6 to 10 others were “moved to an enclosed outdoor environment that was built for them” when their caretakers moved away [86,87].In 1998, the last two known litters of kittens (each three in number) were born on the waterfront. One litter was mothered by an elusive long-term resident cat known as “Miss Witch” [51]; the other litter was produced by a new arrival named “Scarlett” [88]. All six kittens were captured and adopted [51,88].In 2001 after three years of being “kitten-free”, approximately 40 cats remained on Newburyport’s central waterfront [89].October 2002: “We decided to close the feeding station that was close to the Black Cow [formerly the Captain’s Quarters restaurant] (Figure 1, site #6) due to the reduction in feral cat population there” [76].December 2004: due to rumored construction (that did not materialize) at the Windward Boatyard (Figure 1, site #5), 15 elderly cats were adopted by colony caretakers. “After those cats were taken inside, I’m going to guess that there were 25–30 [in total] that remained”, recounted Grimes [90].December 2009: Zorro, an offspring of Miss Witch [91] and the last known cat on the waterfront, died at an estimated age of 16 [92].

It took approximately 2.5 years for MRFRS volunteers to trap and sterilize what was judged to have been all of the cats who lived on the waterfront at the program’s inception [52,68]. Of the initial ~300 cats, it is estimated that two-thirds were returned to the waterfront after sterilization and one-third were put up for adoption [52,67,93]. Over the course of the program, it was surmised that 5–10% of the cats trapped were euthanized due to serious illness, injury, or positive FeLV/FIV test result—most in the early months of the program [50,52]. However, it was estimated that 40 additional cats took up residence on the waterfront due to abandonment or migration from neighboring communities [60] and that a number of litters of kittens were born prior to completion of sterilization efforts [72,73]. Consequently, it was deduced that in aggregate more than 300 cats were trapped, neutered, and either returned, adopted, or euthanized over the entirety of the program [60,94].

### 3.3. Conditions on the Newburyport Waterfront since the Death of Zorro

Since the death of Zorro, in December 2009, it is has been observed that the central waterfront has remained free of resident feral and stray cats [29,52,95]. According to LeBaron, “There aren’t any managed colonies on the waterfront. We don’t have any known strays down there” [95]. “Indeed, we have not had any kittens come in from the Newburyport waterfront area in many, many years now”, added Liz Pease, current MRFRS executive director [96].

Current Newburyport animal control officer (since 2011), Scott Purdie, stated, “I know of past issues with cats on the waterfront; none exist now” [97]. LaRocque concurred: “When I did [animal control for] that city, we had a large population of cats, but MRFRS did a great job of addressing the situation” [98]. “I became a believer in TNR”, she said [99].

Despite the absence of cats in the area since late 2009, according to Pease, a single feeding station has been maintained at Pike Street (Figure 1, site #9) in order to pay homage to what was accomplished on the waterfront, and “so we have a presence [there] in case any cats in need happen along. But, honestly, in my 12 years of involvement, most of the calls we get from the downtown Newburyport area are [for] owned cats that people see outside and assume are in trouble—when really, they are indoor-outdoor [pet] cats” [33].

## 4. Discussion

### 4.1. Factors Contributing to the Elimination of the Waterfront Cats

The free-roaming cat population on Newburyport’s central waterfront was essentially unmanaged for many years before finally reaching crisis proportions in the early 1990s [50,53,82]. Sporadic attempts at lethal control, if they occurred, had been unsuccessful. Per all accounts, it was not until initiation of an intensive TNR campaign, combined with adoption of sociable cats and kittens, that the situation improved. Available evidence indicates that, consistent with the findings of previous studies, the number of free-roaming cats was quickly reduced due to adoption [27], then was further diminished over a number of years (in this case, ultimately to the point of elimination) due to attrition [4,21,22,24].

It is indeterminable whether the waterfront’s free-roaming cat population would have declined similarly had an alternative tactic, such as sustained lethal control, been employed. No attempts to use lethal control could be documented. Evidence points to strong public support for non-lethal management of free-roaming cats within the Newburyport community [50,77], as has been found elsewhere [100,101,102,103], and is likely the reason systematic attempts at lethal control could not be uncovered.

Adoption was thought to be the primary cause of the reported initial reduction in free-roaming cat numbers. Adoption of sociable cats and kittens, which became an essential part of the MRFRS program, has been found to expedite reductions in free-roaming cat numbers [4,21,23,24,27] and is now commonly considered part of TNR program best practices [104]. Essential to MRFRS adoption efforts was the opening of the cats-only limited admission shelter, which reduced the expanding burden on the group’s foster home network and provided a much-needed service for the community at large—admission of unwanted pet cats. Abandonment of pets can be a significant source of free-roaming cats [105,106,107]. The perceived role played by the shelter in reducing the number of free-roaming cats on the waterfront—both by facilitating the adoption of sociable cats and kittens pulled from the area and by providing local cat owners with an alternative to abandonment—exemplifies the interrelatedness of TNR and other animal welfare efforts within a community, as noted by Slater and Shain [31].

The estimated one-third of the cats trapped, sterilized, and subsequently put up for adoption was lower than that experienced at other studied sites where intensive TNR, combined with adoption, produced significant declines in free-roaming cat populations [4,24,27]. Relocation was another source of population reduction, but only for a relatively small number of cats. As previously mentioned, several years into the program, approximately 6 to 10 cats were moved to a specially-built outdoor enclosure when their long-time colony caretakers moved away. This situation was an anomaly and, unlike adoptions, not a regular part of the TNR program [108].

In addition to adoption, particularly in the early months of the TNR program, euthanasia of seriously ill and injured cats, as well as those testing positive for FIV or FeLV was acknowledged to have contributed to the reduction in waterfront cat numbers, though only to a minor degree. It is likely that this factor accounted for no more than 20–25 cats, mostly during the initial phase of the project when cats testing positive for FIV or FeLV were euthanized [52].

Disease is believed to have been a relatively insignificant factor in the long-term decline of the waterfront cat population; in fact, a steady improvement over time in the general health of the cats was observed [50,51]. This trend was attributed, in part, to the disciplined feeding regimen adhered to as part of the MRFRS program, which established a consistent source of nourishment for the waterfront cats [50,109]. Poor nutrition has been associated with greater susceptibility to disease and parasite infestation among free-roaming cats [110]. In addition, as has been documented elsewhere [111,112], improved body condition of the cats after sterilization was noted [50]. Moreover, it is believed that the removal of sociable cats for adoption, which caused an immediate drop in waterfront cat population, combined with the sterilization and vaccination against panleukopenia (as well as rabies) of cats returned to the area after trapping, likely reduced the incidence of illness and injury associated with agonistic behaviors [50] recognized to occur more frequently among unaltered male cats living in high-density populations [113,114]. Female cats likely derived health benefits from the waterfront sterilization efforts as well due to the elimination of physical stresses related to mating, pregnancy, kitten birth, and lactation [24,115,116,117]. As was observed at a site in Central Florida where TNR efforts were monitored for more than a decade [4], it was found that after sterilization and return, many of the waterfront cats lived long, healthy lives, some well into their “teens”. Most of these cats spent their entire lives outdoors, while some, in their later years (as chronicled above), were adopted into homes [60].

Notwithstanding the absence of systematically collected population data, the stakeholders interviewed expressed a very high degree of confidence that the TNR program put in place in 1992 was the principal impetus for the elimination of the area’s estimated 300 free-roaming cats [53,62]. Other potentially contributing factors were dismissed or judged to be incidental [52,75].

### 4.2. Consideration of Other Potentially Contributing Factors

The present investigation appears to have confirmed at least two basic facts: 1) there were many cats, approximately 300, living on the Newburyport waterfront (a general condition that had persisted for many years) in 1992 when the TNR program described above was initiated, and 2) by the end of 2009, no cats remained. Establishment of a causal relationship between TNR and the elimination of the waterfront cats is beyond the capacity of this single descriptive case study, especially considering its lack of associated population data. Nevertheless, the available evidence regarding several possible alternative explanations was carefully considered.

Disease: As stated previously, euthanasia due to detection of serious health concerns was a factor early on, but did not appear to have been a significant long-term cause for the observed population decline. All but a small percentage of cats were reported to be sufficiently healthy to allow for their return to the waterfront or adoption into indoor homes after sterilization. Moreover, no evidence was uncovered via interviews or examination of news accounts and surviving documents that would indicate significant numbers of sick or dying cats among those returned to the area after sterilization and vaccination.

Natural disasters: Interviewees recalled no instances of injury or death to waterfront cats resulting from a natural disaster, such as a hurricane, tornado, or earthquake, even when a list of such events affecting the general area between 1992 and 2009 was provided. Eight hurricanes, or the remnants thereof [118], one tornado [119], and two earthquakes [120], of 3.0 and 3.6 on the Richter scale, were recorded in Essex County during the relevant time period. “I don’t have any recollection of those weather events impacting the cats…the cats and feeders were used to pretty harsh weather conditions”, explained LeBaron [81]. Grimes responded in a consistent manner, but added, “Several severe blizzards and snowstorms would keep [the cats] holed up for a day or so” until colony caretakers could clear paths to their feeding stations [109]. It was speculated by Grimes that some geriatric or unhealthy cats who occasionally disappeared may have succumbed to inhospitable winter weather conditions [109]. The impact of natural disasters on the waterfront’s free-roaming cat population during the examined time period appeared to be inconsequential and was, in any case, likely consistent with what had been occurring for decades prior to TNR efforts beginning.

Changes to the landscape: Much of Newburyport’s transformation from an aging industrial center to a thriving tourist destination took place prior to 1992 [42,43,45]; however, changes to the landscape continued during the time frame of the MRFRS waterfront TNR program. Old outbuildings, shacks, and winch houses, once scattered along the banks of the river, were removed [47,121]; tourist-friendly attractions were added, such as a park, a bicycle/pedestrian trail, and a boardwalk; yacht clubs were expanded; and two large, vacant industrial sites were converted into shops, restaurants, offices, multi-unit housing, and parking [46,47,48,122]. It is unclear what impact, if any, such changes had on the waterfront’s free-roaming cat population; however, it was noted that the bulk of the described disruptions happened after a significant reduction in cat numbers had already occurred. “The cat situation was under control by the time most of the [waterfront] redevelopment took place”, explained Magnanti [53]. Nevertheless, it was speculated by several MRFRS volunteers that ongoing waterfront redevelopment might have played a role in deterring new cats from taking up residence in the area [47,68,121]. Moreover, reduced abandonment of pet cats, associated with the establishment of the cats-only limited admission shelter, and the achievement of 100% sterilization of resident free-roaming cats on the waterfront likely inhibited repopulation. 

## 5. Conclusions

The TNR program instituted in 1992 on Newburyport’s central waterfront is, given its iconic status as one of the most well-known and longest-running TNR campaigns in the U.S., worthy of the present examination. It was a pioneering effort that was sustained over many years in order to achieve its original goal of 100% sterilization of resident cats and zero kitten births along the river [29,50,86]. Moreover, as the foregoing narrative reveals, the program was modified as needed to include a number of innovative tactics that would later become TNR best practices, including pairing TNR with the adoption of sociable cats and kittens, the cultivation of an array of collaborative community partners, and the targeting of intensive trapping and sterilization efforts in an area known to have a high density of free-roaming cats. The use of feeding logs to monitor and record attendance and activity at cat colony feeding sites was another innovative practice incorporated into the program.

Regrettably, because feeding logs and medical records dating back to 2009 and earlier were not preserved, it is likely that valuable data relating to changes in the population of free-roaming cats on the waterfront were lost (though the completeness and specific utility of the lost records are unclear). “Unfortunately, there is very little statistical data for the simple reason that we were so busy trapping, neutering, and releasing cats and caring for non-feral cats we were all fostering in our homes, we had little time to make up stat reports”, explained Fairweather [74].

The lack of consistent sets of systematically collected population data relevant to this case accentuates the importance of the call by Boone and Slater [7,18] and others [24,123] for the professionalization and standardization of TNR data collection and assessment practices. Consistent adherence to an efficient and practicable census conducted at predetermined time intervals is necessary for assessing free-roaming cat population trends and measuring TNR program impacts in a more scientifically robust manner [7].

As described above, some elements of the systematic counting protocol being called for were part of the Newburyport program’s monitoring efforts, though it is frustrating to consider what data might have been collected, preserved, and now available for analysis had heightened awareness concerning the value of such information existed contemporaneously to these efforts. Still, while the Newburyport TNR program underscores the pressing need for broad adoption of systematic data collection and assessment processes, it also highlights the potential effectiveness, feasibility, and desirability of TNR as a management practice.

## Figures and Tables

**Figure 1 animals-07-00081-f001:**
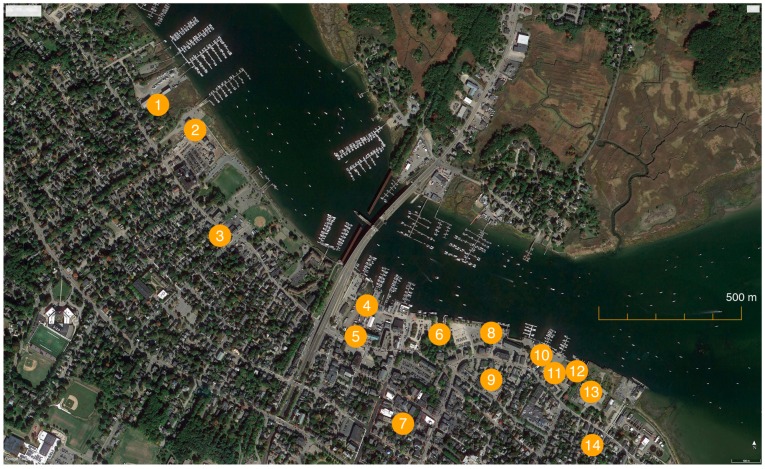
Locations of feeding stations established as part of the trap-neuter-return project for managing unowned, free-roaming cats on the Newburyport, MA waterfront.

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
