# Peer review of "An Examination of an Iconic Trap-Neuter-Return Program: The Newburyport, Massachusetts Case Study"

_animals, 2017, doi:10.3390/ani7110081_

Round 1

Reviewer 1 Report

Given the historical relevance of the Newburyport TNR project, this article is an important contribution to the understanding of the role of TNR programs in community cat population control and in the success of TNR programs. 

The article is well-written, easy to understand and documents, to the best of your ability given the limited data collection/retention, this important TNR program

I found only two sentences that need clarification.

Consider changing the sentence on lines 326 - 329 from "Some of the cats were found indoor homes, while others were 'moved to an enclosed outdoor environment that was built for them' when their caretakers moved away" to "Some of the cats were placed in indoor homes, while others were 'moved to an enclosed outdoor environment that was built for them' when their caretakers moved away."

Consider removing the word "once" from the sentence on line 430 - 440.  The sentence would then read "Most of these cats spent their entire lives outdoors, while some, in their later years (as chronicled above), were adopted into homes."

Author Response

Thank you for your constructive feedback.

The following revisions have been made to the manuscript:

The sentence beginning on line 326 and ending on line 328 has been revised to read “Some of the cats were placed in indoor homes, while 6 to 10 others were ‘moved to an enclosed outdoor environment that was built for them’ when their caretakers moved away.” This change replaces the word “found” with “placed in” as you suggested, and reconciles this sentence with line 412, as instructed by reviewer #2.

The sentence that began on line 419 and ended on line 421 has been removed per the suggestion of reviewer #2. This change resulted in the deletion of citation #109, which caused the order of citations 109-114 to be revised in the reference list.

Per your recommendation, the word “once” has been removed from the sentence beginning on line 438 and ending on line 439. It now reads: “Most of these cats spent their entire lives outdoors, while some, in their later years (as chronicled above), were adopted into homes.”

We hope you find these revisions to be satisfactory.

Thanks again.

Reviewer 2 Report

This article is a descriptive study regarding the TNR program initiated in Newburyport, MA, during the early 1990s.  Although many of the original records regarding this undertaking were no longer available for reference in this study, the authors undertook a substantial amount of effort to collect eyewitness testimony and piece together a timeline and methodology of the project, while also sufficiently examining possible reasons why the program was so successful.  It is a well-written paper, with only minor typographical errors and a few redundant transitional adverbs, which can be easily fixed.  While exact statistical data and permanent records are lacking, this article still presents a useful summary of the practices used, the timeline, and maybe even more important for today’s TNR programs – real-world commentary from those directly involved in the program.  This article is expected to be of great interest for the TNR and shelter medicine population at large.

Revisions suggested are minor and listed below:

1)    Lines 325 and 412: The number of cats relocated to the outdoor enclosure varies significantly between these two lines.  Please reconcile.

2)   Lines 419-422:  There does not appear to be any suggestion within the article that the FIV+ cats euthanized were all male; or that the euthanasia of FIV+ cats was eliminated due to the lack of disease within the population.  This statement should be removed or reworded, as it suggests a finding that is not supported within the article currently.

Overall, a meaningful examination of an early, successful TNR program.

Author Response

Thank you for your constructive feedback.

The following revisions have been made to the manuscript:

The sentence beginning on line 326 and ending on line 328 has been revised to read “Some of the cats were placed in indoor homes, while 6 to 10 others were ‘moved to an enclosed outdoor environment that was built for them’ when their caretakers moved away.” This change replaces the word “found” with “placed in” as suggested by reviewer #1, and reconciles this sentence with line 412 (as you instructed).

The sentence that began on line 419 and ended on line 421 has been removed per your suggestion. This change resulted in the deletion of citation #109, which caused the order of citations 109-114 to be revised in the reference list.

Per reviewer #1, the word “once” has been removed from the sentence beginning on line 438 and ending on line 439. It now reads: “Most of these cats spent their entire lives outdoors, while some, in their later years (as chronicled above), were adopted into homes.”

We hope you find these revisions to be satisfactory.

Thanks again.